# The Identification and Clinical Applications of Mutated Antigens in the Era of Immunotherapy

**DOI:** 10.3390/cancers14174255

**Published:** 2022-08-31

**Authors:** Yae Kye, Lokesh Nagineni, Shrikanth Gadad, Fabiola Ramirez, Hannah Riva, Lorena Fernandez, Michelle Samaniego, Nathan Holland, Rose Yeh, Kei Takigawa, Subramanian Dhandayuthapani, Jessica Chacon

**Affiliations:** 1Paul L. Foster School of Medicine, Texas Tech University Health Sciences Center El Paso, El Paso, TX 79905, USA; 2L. Frederick Francis Graduate School of Biomedical Sciences, Texas Tech University Health Sciences Center El Paso, El Paso, TX 79905, USA; 3Center of Emphasis in Cancer, Department of Molecular and Translational Medicine, Texas Tech University Health Sciences Center El Paso, El Paso, TX 79905, USA; 4Mays Cancer Center, UT Health San Antonio MD Anderson Cancer Center, San Antonio, TX 78229, USA; 5Center of Emphasis in Infectious Diseases, Department of Molecular and Translational Medicine, Texas Tech University Health Sciences Center El Paso, El Paso, TX 79905, USA

**Keywords:** mutated antigen, neoantigen, immunotherapy, tumor-associated antigens

## Abstract

**Simple Summary:**

In this review article, we highlight the process and importance of identifying neoantigens for immunotherapy for cancer. Although the process of identifying neoantigens may be time-consuming and costly, various efforts are being conducted at different stages of pre-clinical and clinical development to improve the process and identify neoantigens. This initiative will be imperative in developing the next generation of cost-efficient and potent immunotherapies against cancer.

**Abstract:**

The era of personalized cancer therapy is here. Advances in the field of immunotherapy have paved the way for the development of individualized neoantigen-based therapies that can translate into favorable treatment outcomes and fewer side effects for patients. Addressing challenges related to the identification, access, and clinical application of neoantigens is critical to accelerating the development of individualized immunotherapy for cancer patients.

## 1. Introduction

Immunotherapy has rapidly become a primary mode of cancer treatment due to the remarkable success and clinical improvement seen in patients with advanced-stage or aggressive, recurrent cancers such as melanoma, ovarian cancer, breast cancer, and gastrointestinal cancers [1]. A significant number of therapies that have seen success to date involve the utilization of T cells. This is because of the tumor cell’s unique ability to suppress the immune system by modulating its surrounding microenvironment, also known as the tumor microenvironment (TME). By targeting the TME with antigen-specific immune cells, we are able to essentially activate the immune system to mount an anti-cancer attack.

There has been much interest in identifying, developing, and producing immune cells and proteins that are able to mount a potent immune response against the tumor cells and their TME. This vested interest has led to the production of various immunotherapies including monoclonal antibodies (mAbs), tumor-infiltrating lymphocytes (TIL), and chimeric antigen receptor T-cells (CAR-T). Despite these advancements, due to the metabolic reprogramming and adaptability of cancer cells, these immunotherapies have been met with both innate and acquired resistance [2,3]. This has led to the search for new immunotherapy targets and has opened a new hallmark of cancer immunotherapy.

Neoantigens are tumor-specific proteins synthesized by tumor cells as protein by-products [4,5]. The rapid division and proliferation of tumor cells lead to various mutations in coding and non-coding loci [4,5]. The changes in the amino acid sequence that occur due to mutations in coding regions leads to the production of proteins that are not found in normal cells and are unique to tumor cells [4,5,6]. These tumor-specific by-products termed *neoantigens* are unique in that multiple patients can share the same neoantigens (shared neoantigens), but they can also be specific to an individual (personalized neoantigens) [4,5,6]. In addition, neoantigens are highly immunogenic, making them a favorable immunotherapy target [5,7]. Thus, the identification of neoantigens and the development of therapies targeting them highlights a promising field that may yield the next generation of cancer treatments and advance the field of personalized medicine. This review will focus on the current available immunotherapies as well as the identification, manufacturing, and clinical applications of neoantigens.

## 2. Cancer and Immunotherapy

The relationship between cancer and the immune system is very dynamic. Cancer cells modulate the immune system to survive, proliferate, and metastasize. For instance, tumor cells express immuno-suppressive ligands on the cell surface such as programmed death ligand-1 (PDL-1) to evade immune cell recognition and inhibit the mounting of an immune response. In addition, tumor cells can also hijack and employ crosstalk to mediate immune cells and inflammation to aid with metastasis [8,9]. The basis of immunotherapy is to target these specific areas of crosstalk, intervene, and modulate the immune response to target the cancer cells. The ultimate goal of immunotherapy is to enhance recognition, target, and mount a toxic response against the cancer cells [4].

Monoclonal antibodies (mAbs) are a type of targeted immunotherapy that has shown promising therapeutic results that is currently being used to treat various cancers including breast cancer, colorectal cancer, and leukemias [5,7,10]. For instance, mAbs can mount a potent immune response and can be employed to trigger cytotoxicity and inhibit further tumorigenesis (Figure 1A) [1,11,12]. They are able to recognize both cell surface antigens and secreted antigens with high specificity, indicating their versality [10].

One of the more successful therapeutic mAbs are immune checkpoint inhibitors (ICIs) [4] This is possible because of the T cells’ unique ability to moderate their own immune response. T cells keep the immune system from becoming overly active and destructive. This self-modulating mechanism is conducted by immune checkpoint proteins such as CTLA-1 and PD-1, where the binding of the respective receptors and ligands on T cells induces a regulatory response and curtails the immune response [13,14]. Cancer cells take advantage of this self-regulating function of T cells by synthesizing and presenting these markers to the cell surface, inhibiting the T cell from mounting a response, allowing the cancer cell to evade apoptosis and survive [1,11,12]. ICIs target these immunosuppressive ligands and receptors and renders them inactive, thereby turning on and activating the immune system (Figure 1B). By targeting these immune checkpoint markers, cancer cells are unable to regulate or evade the immune system, making them susceptible to apoptosis. Retrospective studies have shown that ICIs, such as the CTLA-4 mAb, ipilimumab, are effective for the treatment of metastatic melanoma [15,16,17,18]. Additionally, the combination of anti-PD-1/PD-L1 and anti-CTLA-4 mAbs has been shown to be therapeutically responsive in treating various cancers including hepatocellular carcinoma and ovarian cancer [19,20,21,22].

Adoptive cell therapy (ACT) is a passive immunotherapy that uses tumor-infiltrating lymphocytes (TIL) to induce tumor-suppressive or cytotoxic effects. ACT entails isolating and extracting immune cells from patients, inducing cell differentiation ex vivo, and expanding and re-infusing the cells back into the patients (Figure 1C). It is used to grow a large army of immune cells outside the body that when re-infused back into the patient lead a potent immune response against antigen-specific tumor cells [6,7]. TIL therapy has shown significant results in the treatment of several cancers, especially metastatic melanoma [15,16,17,18,22]. One of the major advantages of utilizing TILs for immunotherapy is the diverse repertoire of antigens that TILs can recognize.

Other ACTs include T-cell Receptor-engineered T cell (TCR-T) and Chimeric Antigen Receptor T-cell (CAR-T) therapies. TCR-T therapy uses genetically modified natural T cells to target and destroy tumors and relies on the interaction between the peptide-MHC complex to mount a potent cytotoxic response against the tumor. TCR-T has been used to target various tumors and hematologic malignancies and current studies have shown TCR-T to be more effective against solid tumors rather than hematologic malignancies [7,23]. In contrast, CAR-T therapy uses genetically engineered T cells that can mount a cytotoxic response without the peptide–MHC complex interaction (Figure 1D) and has shown clinical efficacy against hematologic malignancies, with efficacy against solid tumors currently being thoroughly investigated [24,25].

Despite significant increases in overall survival and the therapeutic success found with these immunotherapies, they are not without adverse effects or complications. Immunotherapeutic agents have led to the development of serious (and sometimes fatal) immune-related adverse events (irAEs) in a subset of patients [26]. Therapeutic response rates have also varied among patients and different solid tumor types [14,27] In addition, resistance to these immunotherapies is an unfortunate complication that limits the duration and efficacy of treatment. Although there have been various methods to combat these barriers including the use of a combination of mAbs to induce a polyclonal response, there still exists room for optimization and further development [1,11,12,24,25]. With the advancements in high-throughput sequencing, prediction algorithms, and screening and characterization technology, the focus of immunotherapy has turned towards neoantigens.

## 3. Neoantigens

Tumor antigens can be divided into three different categories: Tumor Associated Antigens (TAAs), Cancer Testis Antigens (CTAs), and Tumor Specific Antigens (TSAs). TAAs are considered self-antigens that are expressed in both normal and cancer cells. These TAAs are abundantly expressed in cancer cells and can elicit an anti-tumor immune response. CTAs are another type of tumor-specific antigens that are only expressed in germline cells. Developing therapies against CTAs thus allows us to target germline antigens without damaging the somatic cells [28,29]. Recent progress in cancer therapy has focused on targeting TSAs. TSAs are foreign, typically arising from genomic mutations or viruses, and thus are not considered self-antigens [4,5,6,30]. They are unique to the tumor genome, making them excellent therapeutic targets. Targeting TSAs allows us to attack the cancer cells specifically without affecting normal cells.

Neoantigens are a type of TSA. They are unique protein by-products that arise from the genetic instability and aberrant mutations of tumor cells [5,6,7,8]. These neoantigens are specific to cancer cells and are not found in normal tissue. They vary from patient to patient and can also be found across multiple cancers [4,5,6,7]. Neoantigens that are unique to the individual cancer patient are referred to as *personalized neoantigens.* Targeting these personalized neoantigens leads to the development of personalized therapy. On the other hand, neoantigens can also be found across multiple cancer patients that are not found in the normal genome and are known as *shared neoantigens* (Figure 2). Shared neoantigens play a role in driving tumor growth and consist of most major oncogenes, including BRAF mutations in melanoma and KRAS mutations in pancreatic, colorectal, and endometrial cancers [5,6,7,13]. Additionally, these molecules are less prone to being lost as targets due to the essential nature of their mutations and imperative role in tumor growth and metastasis. The development of therapies against shared neoantigens is favorable as they hold the potential to treat a broad population of patients [4].

## 4. Identification of Neoantigens

The current methodology of identifying neoantigens includes an exhaustive list of sequencing, quality control checks, realignments of tumor and normal sequencing data, and comparing the normal and tumor alignments to determine somatic mutations [31]. These somatic mutations must then be converted to their corresponding mutated peptide sequences and undergo human leukocyte antigens (HLA)-allele typing [31]. Once the HLA-allele and mutated epitope affinity is assessed using prediction algorithms such as netMHCpan, it is henceforth called a neoantigen. Afterwards, the ability of these neoantigens to be recognized by immune cells are determined by running various immunological assays [31,32].

There are multiple areas within these steps that require quality control including the maintenance and handling of the tumor samples and the sequencing and alignment data. The purity of the tumor samples must be maintained as it can easily decrease the sensitivity of the somatic mutation calling [31,32]. In addition, there are multiple methods and algorithms that are used to help optimize and lower the false positive rates with the sequencing and alignments [31]. The most important step in this multi-step process is the prediction of the HLA binding capacity to the mutated peptide. These prediction algorithms assess and rank the HLA-peptide by their binding affinity which corresponds to the neoantigen’s ability to be recognized by the immune system [31,32,33].

Though neoantigens are tumor specific, immunogenic, and have shown good clinical results, the proportion of neoantigens with immunological significance is surprisingly low [4,31]. In addition, the identification of these immunogenic neoantigens is laborious, costly, and time-consuming, highlighting one of the challenges in developing therapies using personalized neoantigens. This limitation can be addressed by targeting shared neoantigens which are known to be broadly recurrent and immunogenic amongst cancer patients [31].

## 5. Clinical Applications of Neoantigens

Once a neoantigen is identified and its immunological significance has been verified, it can be applied in various clinical scenarios including diagnostic and treatment screening and various immune-based therapies.

## 6. Neoantigen-Based Biomarkers

Multiple studies have investigated the relationship between neoantigens and disease progression and have shown that a higher neoantigen load is associated with better overall survival in cancer patients [6,34]. Furthermore, studies have also shown metastatic progression occurring with the loss of neoantigens [6]. These findings suggest the possible role of neoantigens as biomarkers of tumor burden and disease progression as well as patient survival, adding to the growing interest in identifying neoantigens. Currently, there are nine actively recruiting clinical trials dedicated to identifying neoantigens across various cancers and patient populations (Table 1).

## 7. Neoantigen-Based Therapies

As mentioned, tumor neoantigens are specific to the tumors of individual patients and are highly immunogenic making them a favorable target in cancer therapy [29,35]. They also are highly specific, have numerous targets, and have broad efficacy [35]. By capitalizing on new sequencing technologies that help identify particularly relevant neoantigens, immunotherapy agents including neoantigen-specific immune cells and neoantigen-based vaccines can be further developed and optimized for clinical potency [7,36].

### 7.1. Neoantigen-Specific TCR-T Cell Therapy

An extension of neoantigen specific T cells is TCR-T cell therapy. It is an adaptation of the current TCR-T cell therapy, where a TCR is genetically engineered to be specific for the identified neoantigens. These engineered TCR T cells (TCR-T) targeting tumor-specific antigens are expanded and infused into the patient as a form of therapy. This therapy combines the genetic engineering of TCRs and the sequencing and mutation alignment algorithms in identifying neoantigens to create a synthetic TCR-T targeting individual or multiple tumor specific antigens.

Neoantigen-specific T cells can be seen as an adaptation to ACT. The idea is to essentially build a large army of T-cells that recognize and bind the patient or tumor-specific neoantigens to mount a cytotoxic response against the tumor. Once a neoantigen is identified, autologous T cells from the patient are obtained and undergo stimulation by neoantigen presentation by APCs. The stimulated T cells are then isolated, expanded, and infused back into the patient (Figure 3A). These neoantigen-specific T cells are expected to recognize the same neoantigens and launch a cytotoxic immune response against the tumor.

The development of neoantigen-specific T cells is a multi-step process that requires the meticulous handling of samples and expertise in computational analysis (Figure 3A). First, samples from the tumor and normal tissue or blood are taken from the patient. This is then followed by genomic sequencing and somatic mutation calling followed by sequence comparison and alignment. Once the tumor-specific mutations are identified, exome and transcriptome sequencing and computational screening are performed to create synthetic neoantigen DNA, mRNA, or peptides. These synthetic neoantigens are introduced to APCs, where they are processed and mounted onto their respective HLAs. Following the identification of neoantigens, T cells are also isolated and enriched for antigen-specific T cells in lieu of bulk T cells. These cells undergo antigen stimulation by the processed neoantigen-HLA bound APCs. In order to identify the stimulated T cells, co-stimulatory molecules, such as CD137, are screened using flow cytometry to ensure antigen-specific T cell activation [13,17,18,22]. The expression of CD137 is transient and can be seen on CD4+ and CD8 + T cells. Therefore, it is imperative to investigate the role of antigen-specific T cells in both CD4+ and CD8+ subsets. Once antigen-specific T cell activation has been confirmed, the activated T cell populations are collected and expanded in vitro. This subset of T cells is neoantigen-specific TCR-T cells. These autologous T cells are infused back into the patient where they will launch an immune response upon recognition of the neoantigen.

Currently, there are nine active clinical trials involving neoantigen-specific TCR-T cell therapy and one study that has been terminated (Table 2). In addition, there are two actively recruiting neoantigen-specific TCR-T cell clinical trials (Table 2). These TCR-T therapies have been on the rise and have shown promising therapeutic results against cancers particularly in ovarian cancer and epithelial cancer [37,38,39].

### 7.2. Neoantigen Vaccines

Historically, cancer vaccines were developed to target TAAs. However, TAA vaccines met limited therapeutic success as they were often unable to elicit a potent immune response to avoid autoimmunity [2]. This can be attributed to the lack of tumor specificity as TAAs are present in both normal and tumor cells [28,29]. Neoantigen-based vaccines are more beneficial than TAA-based vaccines as neoantigens are expressed solely by tumor cells [40]. The unique expression pattern of neoantigens can be utilized to avoid autoimmunity and mount a potent immune response possibly leading to immunological memory development and protection against tumor recurrence [7,12,36].

Neoantigen-based vaccines are composed of immunogenic neoantigens that are designed to ultimately induce de novo cytotoxic T cell responses [7,29,35]. Once the vaccine is administered, the neoantigens are processed and presented to T cells by APCs. Once recognized, the T cells will become activated and subsequently mount a cytotoxic response and infiltrate into the tumor cells, turning “cold” tumors into “hot” tumors [27,41]. *Cold tumors* are tumors that generally do not respond to immunotherapy, while *hot tumors* usually respond to immunotherapy due to the infiltration of T cells and pro-inflammatory cytokines [27,41].

The development and manufacturing of neoantigen vaccines is a multi-step and laborious process (Figure 3B). There are various types of neoantigen cancer vaccines that have been developed including peptide vaccines, mRNA vaccines, DNA vaccines, viral vaccines, and dendritic cell (DC) vaccines [30,42]. The first step involves obtaining a biopsy of tumor tissue and normal tissue or peripheral blood samples. These samples are then screened and compared to identify and verify tumor-specific mutations by exome sequencing. This is then followed by sequence alignment and computational analysis to predict the affinity of potential neoantigens to respective HLAs. Once the prediction algorithms rank the neoantigen-HLA binding affinity and neoantigen candidates are selected, vaccine design and development ensues and finally administered to the patient. There is not a set schedule for vaccine administration; they seem to differ based on vaccine type, cancer type, response rate, and the patient.

There have been multiple clinical studies that developed and administered neoantigen vaccines to treat cancers, including gastrointestinal cancers, melanoma, and nasopharyngeal cancers, with promising results [38,43,44]. Previous studies found that neoantigen vaccines are best used in cancers with high rates of somatic mutations and vaccines with a larger load of specific neoantigen peptides increased the potential of a favorable clinical response [4,5,6,30]. In addition, a prior study used genomic and bioinformatics to identify shared neoantigens in patients with refractory solid tumors and demonstrated a durable response and stabilization of the diseases [45]. Currently, there are 71 clinical trials that have utilized or are utilizing neoantigen-based vaccines to treat multiple neoplasms (Table 3). Personalized neoantigen vaccines and/or neoantigen vaccines in conjunction with other immunotherapies are being studied in various cancers including small cell lung cancer, pancreatic cancer, kidney tumors, and more broadly solid tumors to determine its efficacy and therapeutic potential (Table 3).

Though prior studies have shown promising results, these current ongoing trials will help further our understanding of neoantigen and its vaccines including its safety, duration of efficacy, and more. These studies also highlight a quickly advancing niche in cancer immunotherapy that can lead to the production of numerous immunotherapies against a wide array of cancers and personalized medicine.

## 8. Limitations and Challenges

There are several limitations to the identification of neoantigens and the implementation of neoantigen-based therapies in the clinical setting. First, tumor mutational burden may affect the effectiveness of such therapies, particularly neoantigen-targeted vaccines. For example, in a study looking into the identification and the use of mutations for neoantigen-targeted vaccines, a total of 962 mutations were reported [23]. Given that most human tumors contain an average of two to six mutations, the utility of neoantigen-targeted vaccination is questionable for tumors with low mutational burden [37,38].

The identification of neoantigens and biomarkers has been made possible by the emergence of techniques such as next-generation sequencing (NGS) and other high-throughput molecular technologies [45]. However, performing the large-scale identification of mutations can present a health care barrier for subsets of the population. For example, studies have found that compared with non-Hispanic whites, Black and Hispanic individuals were less likely to have had NGS testing, despite the Medicare national coverage determination [46,47,48]. Although more research is needed on this subject, these findings suggest that the identification of neoantigens is not equitable, paving the way for health care disparities in the field of immune-targeted therapy.

While TIL therapy is highly effective, the isolation and expansion of mature T cells from the TME increases the risk of terminal differentiation, which may limit the therapeutic effect when reintroducing the cells to the patient [16,49]. The optimal differentiation stage of the T cells, as well as the identification of the T cell receptors that recognize the neoantigens is currently being explored in both CD4+ and CD8+ subsets [7,13,28,50]. In addition, the majority of neoantigens that have been identified are restricted to MHC-I, whereas the role of MHC-II and CD4+ T cells is still being explored [6].

Another technical obstacle related to the creation of neoantigen-targeted therapies, is the identification and prioritization of neoantigens that are present on the cell surface of HLAs. Thus, optimizing the identification of immunogenic neoantigens is of great priority. Additionally, the accuracy in predicting the immunogenicity of tumor neoantigens is another limiting factor. Scientists are attempting to address this challenge by coming up with unique ways of computationally predicting which neoantigens are of interest [39].

The cost of neoantigen research as well as the time for development also presents as a limiting factor. As neoantigens are specific to each person, the treatment is highly personalized. This translates into a costly and time-consuming endeavor. For example, the timing of the development of neoantigen vaccines has been a deterrent, particularly with the pressure on laboratories for production [7,36]. Similarly, from the identification of gene mutations and neoantigens to the validation and production of the neoantigen-based treatment, the cost of personalized neoantigen treatment can be very expensive [7,36]. One strategy for cost reduction may include the development of vaccines targeted at shared neoantigens; this would also potentially significantly increase access of neoantigen vaccines to an appropriately applicable population [29,37,51]. Another strategy to reduce cost is the development of combination therapies of traditional immunotherapy drugs with neoantigen based therapies [7,36]. The development of a more improved and streamlined process for neoantigen-based therapy production to reduce labor and cost as well as the continued analysis of large cancer databases to identify shared neoantigens are some challenges and advances this field will have to address moving forward.

Last but not least, although the developments in immunotherapeutic agents have been promising and have advanced the field of cancer immunotherapy, the long-term safety and efficacy of these agents have yet to be thoroughly investigated. The irAEs, unknown clinical adverse events, rate of recurrence to remission, and rate of morbidity and mortality are possible limitations to the widespread utilization of neoantigen-based immunotherapies.

## 9. Conclusions

The field of immunotherapy has revolutionized the treatment of cancer. In recent years, neoantigen-based therapies have provided patients with alternative treatment options to whom standard immunotherapies were unable to be used or rendered ineffective. Although vaccine development against neoantigens is time-consuming, complex, and expensive, the potential to offset resistance, target specific cancers, and most importantly, personalize therapy holds immense clinical value and merit. The continued optimizations in engineering neoantigen-based therapies and continued investigations of its therapeutic efficacy and safety in clinical trials will be imperative in developing the next generation of cost-efficient and potent therapies against the ever-evolving battle against cancer.

## Figures and Tables

**Figure 1 cancers-14-04255-f001:**
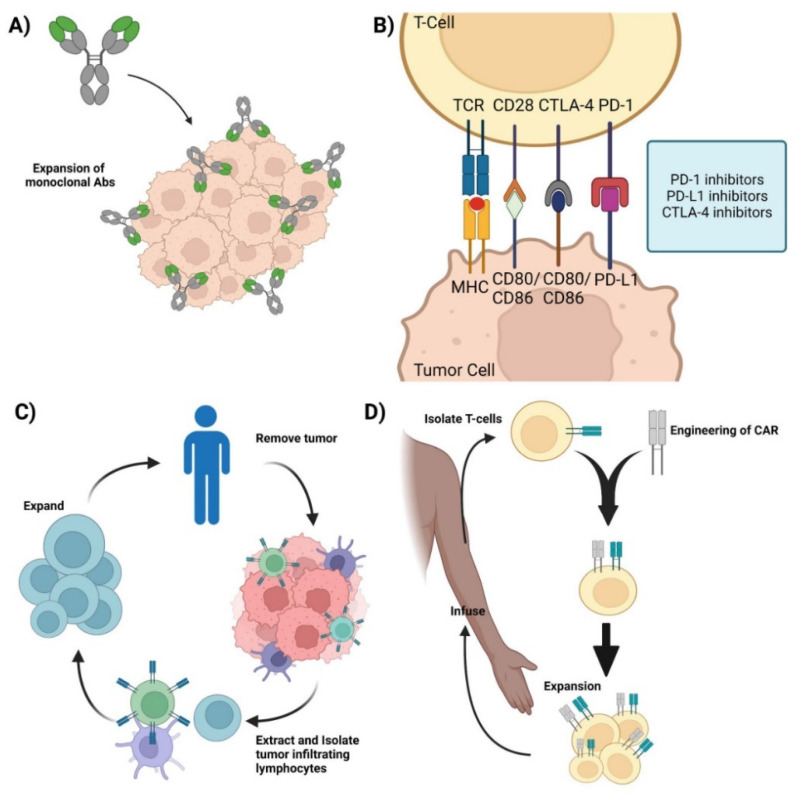
Current immunotherapies. (**A**) *Monoclonal antibodies*—representation of mAbs and the mechanism involved in mounting an immune response against cancer cells. (**B**) *Immune checkpoint inhibitors*—Graphic showcasing the immune crosstalk that occurs between T cells and tumor cells and the respective ligands and receptors targeted using mAbs. (**C**) *Adoptive Cell Therapy*—Graphic showing how TILs are synthesized by removing the respective tumor from the patient, isolating, extracting, and expanding the appropriate TILs, and its re-infusion back into the patient for therapy. (**D**) *Chimeric Antigen Receptor T-cell therapy*—Representation of CAR-T synthesis and production. Isolate T cells from the patient and mount a synthetic CAR onto the host T cells. Expand the modified T cell population and re-infuse back into the patient. Credits: Images were created using BioRender.

**Figure 2 cancers-14-04255-f002:**
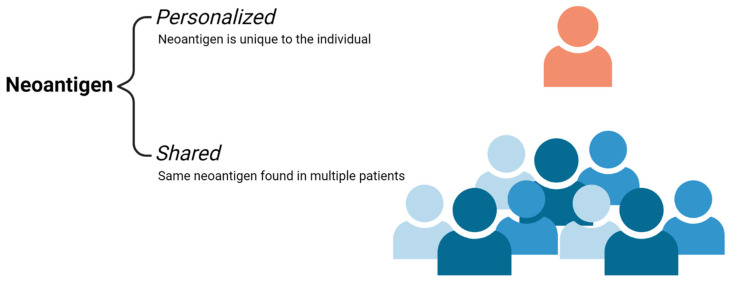
Neoantigen Subtypes. *Personalized* neoantigens are antigens secreted from the tumor that is unique to the individual. *Shared* neoantigens are tumor unique antigens that are found across multiple individuals or a subset of a patient population. Credits: Images were created using BioRender.

**Figure 3 cancers-14-04255-f003:**
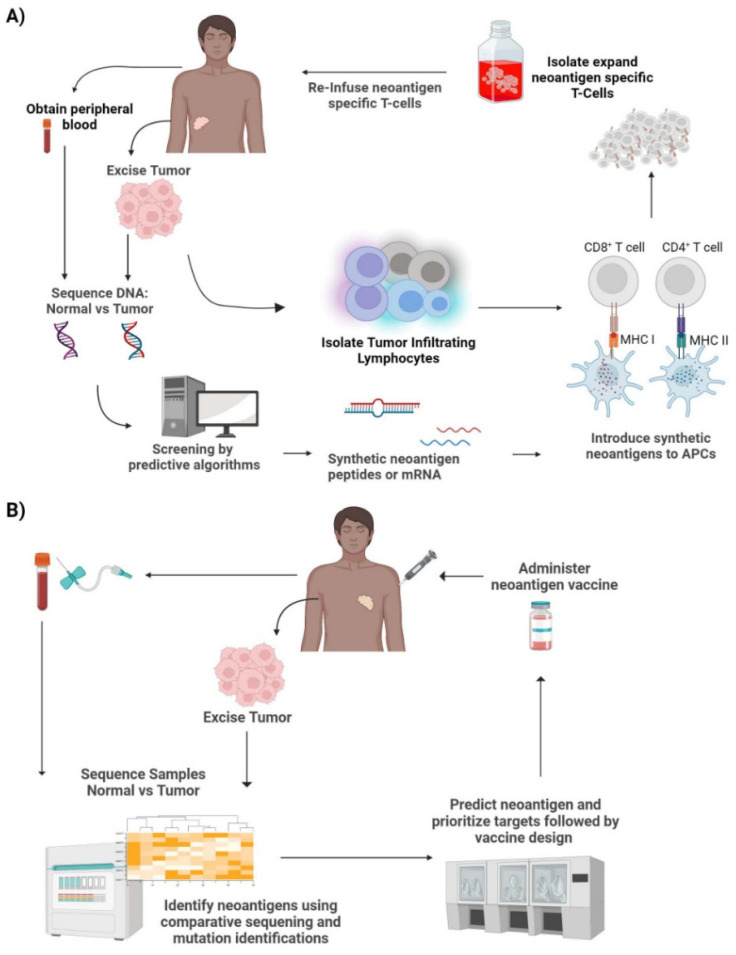
Neoantigen therapies. (**A**) *Neoantigen specific T-cell therapy*—Tumor cells biopsied from the patient can be used for exome/transcriptome sequencing and computational screening to create synthetic neoantigen DNA, mRNA, or peptides. These antigens are processed by APCs and presented to T cells leading to T cell activation. The activated T cells with TCRs specific to the neoantigens are then isolated and expanded. This is then reinfused back into the patient. (**B**) *Neoantigen vaccines*—Tumor cells excised from the patient undergo comparative sequencing and mutation identification with the patient’s normal cells. This is then used to predict neoantigens and prioritize targets. This is followed by vaccine design, production, and finally administered to the patient. Credits: Images were created using BioRender.

**Table 1 cancers-14-04255-t001:** Clinical Trials Identifying Neoantigens.

Cancer	NCT No.	Status	Phase	Number of Patients Enrolled	Intervention
Non Small Cell Lung Cancer, Colorectal Cancer, Gastroesophageal Adenocarcinoma, Urothelial Carcinoma, Pancreatic Ductal Adenocarcinoma	NCT03794128	Completed	Observational	93	Not Applicable
Ovarian Cancer, Endometrial Cancer, Colorectal Cancer, Cholangiocarcinoma, Non Small Cell Lung Cancer, Pancreatic Cancer	NCT05124743	Recruiting	Observational	2000	Not Applicable
Metastatic Colorectal Cancer, Stage II/III Colon Cancer	NCT05158621	Recruiting	Observational	100	Not Applicable
Colon Cancer	NCT04994093	Recruiting	Observational	100	Not Applicable
Non-small Cell Lung Cancer	NCT04923802	Recruiting	Observational	400	Not Applicable
Hematological Malignancies B	NCT02844491	Terminated	Observational	28	Not Applicable
Solid Tumor	NCT03517917	Recruiting	Observational	400	Not Applicable
Synovial Cell, Colorectal, Lung, Bladder Cancer, Melanoma	NCT00001823	Recruiting	Observational	7000	Not Applicable
Acute Myeloid Leukemia	NCT03789981	Recruiting	Observational	75	Not Applicable
Pancreatic Cancer	NCT02869802	Recruiting	Observational	190	Not Applicable
Pancreatic Ductal Cancer	NCT02750657	Active, not recruiting	Observational	332	Not Applicable

**Table 2 cancers-14-04255-t002:** Clinical Trials with Neoantigen specific TCR-T cell therapy.

Cancer	NCT No.	Status	Phase	Number of Patients Enrolled	Intervention
Malignant Epithelial Tumors, Malignant Solid Tumors	NCT05141474	Recruiting	Phase 1, early	10	Single Arm: NEXTGEN-TIL (Neoantigen-based TIL)
Advanced Non- Small Cell Lung Cancer	NCT04032847	Recruiting	Phase 1,2	50	Group 1) ATL001 (autologous clonal neoantigen reactive T cells) Group 2) ATL001 + Pembrolizumab
Solid Tumor	NCT03970382	Active, not recruiting, suspended	Phase 1	21	Sequential: NeoTCR-P1 adoptive cell therapy + nivolumab + IL-2
Endocrine/Neuroendocrine, Non-Small Cell Lung Cancer, Breast Cancer, Gastrointestinal/Genitourinary Cancers, Ovarian Cancer	NCT04102436	Recruiting	Phase 2	210	Single Arm: Fludarabine + Cyclophosphamide + Aldesleukin + Sleeping Beauty Transposed PBL
Melanoma, Non-small Cell Lung Cancer, Squamous Cell Carcinoma of Head and Neck, Urothelial Carcinoma, Renal Cell Carcinoma, Small-cell Lung Cancer, Cutaneous Squamous Cell Carcinoma, Anal Squamous Cell Carcinoma, Merkel Cell Carcinoma	NCT04596033	Terminated	Phase 1	49	Group 1) GEN-011 (T cell therapy) + IL-2 Group 2) GEN-011 + IL-2 + Fludarabine + Cyclophosphamide
Gastrointestinal (GI) Neoplasms, GI Epithelial Cancer, Colorectal, Pancreatic, Gall Bladder, Colon, Esophageal, Stomach Cancer	NCT04426669	Recruiting	Phase 1, 2	20	Sequential: Cyclophosphamide + Fludarabine + TIL + Aldesleukin
Unresectable Melanoma Metastatic Melanoma	NCT04625205	Recruiting	Phase 1	52	Single Arm: NEO-PTC-01 (autologous personalized T cell)
Melanoma	NCT03997474	Recruiting	Phase 1, 2	40	Single Arm: ATL001 (autologous clonal neoantigen reactive T cells) + Checkpoint Inhibitor
Gynecologic Cancer, Colorectal Cancer, Pancreatic Cancer, Non-small Cell Lung Cancer, Cholangiocarcinoma, Endometrial Cancer, Ovarian Cancer, Neoplasm; Lung Adenocarcinoma, Lung Squamous Carcinoma, Lung Adenosquamous carcinoma	NCT05194735	Recruiting	Phase 1,2	180	Sequential: Neoantigen specific TCR-T cell drug product + Aldesleukin (IL-2)
Malignant Epithelial Neoplasms	NCT04520711	Recruiting	Phase 1	24	Single Arm: TCR-transduced T cells + CDX-1140 + Pembro

**Table 3 cancers-14-04255-t003:** Clinical Trials with Neoantigen Vaccines.

Cancer	NCT No.	Status	Phase	Number of Patients Enrolled	Intervention
Non Small Cell Lung Cancer	NCT04397926	Recruiting	Phase 1	20	Single Arm: Individualized neoantigen peptides vaccine
Triple Negative Breast Cancer	NCT04105582	Completed	Phase 1	5	Single Arm: Neo-antigen pulsed Dendritic cell
Non Small Cell Lung Cancer	NCT04487093	Recruiting	Phase 1	20	Group 1) neoantigen vaccine + EGFR-TKI; Group 2) neoantigen vaccine + anti- angiogenesis
Pancreatic Cancer	NCT03645148	Completed	Phase 1	7	Single Arm: iNeo-Vac-P01 (neoantigen peptides)+ GM-CSF
Gastric Cancer, Esophageal Cancer, Liver Cancer	NCT05192460	Recruiting	Not applicable	36	Single Arm: neoantigen tumor vaccine + PD-1/L1
Pancreatic Cancer	NCT03122106	Active, not recruiting	Phase 1	15	Single Arm: personalized neoantigen DNA vaccine
Resectable Pancreatic Cancer	NCT04810910	Recruiting	Phase 1	20	Single Arm: iNeo-Vac-P01 (neoantigen peptides)
Resectable Esophageal Cancer	NCT05307835	Recruiting	Phase 1	40	Group 1) iNeo-Vac-P01; Group 2) GM-CSF
Pancreatic Cancer	NCT03956056	Active, not recruiting	Phase 1	12	Single Arm: Neoantigen Peptide Vaccine + Poly ICLC
Neoplasms	NCT05475106	Recruiting	Phase 1	100	Single Arm: Multi-peptide neoantigen vaccine
Neoplasms	NCT04509167	Completed	Phase 1	30	Single Arm: Neoantigen peptides
Extensive-stage Small Cell Lung Cancer	NCT04397003	Recruiting	Phase 1	27	Single Arm: Carboplatin + etoposide + durvalumab + polyepitope neoantigen DNA vaccine
Pancreatic Cancer	NCT03558945	Recruiting	Phase 1	60	Group 1) Radical surgery and post-operative chemotherapy + personalized neoantigen vaccine; Group 2) Radical surgery and conventional post-operative chemotherapy
Triple Negative Breast Cancer	NCT03199040	Active, not recruiting	Phase 1	18	Group 1) Neoantigen DNA vaccine + Durvalumab; Group 2) Neoantigen DNA vaccine
Metastatic Hormone-Sensitive Prostate Cancer	NCT03532217	Completed	Phase 1	19	Single arm: PROSTVAC + ipilumumab + nivolumab neoantigen DNA vaccine
Colonic neoplasms	NCT05456165	Recruiting	Phase 2	142	Group 1) GRT-C901 (adenovirus vector) vaccine, GRT-R902 (self-amplifying mRNA vector) vaccine + Atezolizumab + Ipilimumab; Group 2) Adjuvant Chemotherapy
Advanced Malignant Solid Tumor	NCT03662815	Active, not recruiting	Phase 1	30	Single Arm: iNeo-Vac-P01 + GM-CSF
Kidney Cancer	NCT02950766	Recruiting	Phase 1	19	Single Arm: NeoVax (Poly-ICLC and Neoantigen Peptide) + Ipilimumab
Advanced Malignant Solid Tumor	NCT04864379	Recruiting	Phase 1	30	Group 1) RFA+PD-1+iNeo-Vac-P01; Group 2) RFA+iNeo-Vac-P01+PD-1
Colorectal neoplasms	NCT05141721	Recruiting	Phase 2, Phase 3	665	Group 1) GRT-C901/GRT-C902 (neoantigen vaccine) + Fluoropyrimidine + Bevacizumab + Oxaliplatin + Atezolizumab + Ipilimumab; Group 2) Fluoropyrimidine + Bevacizumab + Oxaliplatin
Non Small Cell Lung Cancer, Colorectal Cancer, Pancreatic Cancer, Solid Tumor, Shared Neoantigen-Positive Solid Tumors	NCT03953235	Recruiting	Phase 1, 2	144	Sequential: GRT-C903 + GRT-R904 + ipilimumab + nivolumab
Non Small Cell Lung Cancer, Colorectal Cancer, Gastroesophageal Adenocarcinoma, Urothelial Carcinoma	NCT03639714	Active, not recruiting	Phase 1,2	214	Sequential: GRT-C901 + GRT-R902 + nivolumab + ipilimumab
Colorectal Cancer	NCT01885702	Active, not recruiting	Phase 1, 2	25	Group 1: Colorectal Cancer pts with DC Vaccine (Frameshift-derived neoantigen loaded onto DC); Group 2: Lynch syndrome pts with DC vaccine
Anatomic Stage IV Breast Cancer, Clinical Stage III, IV Cutaneous Melanoma, Hormone Receptor-Positive Breast Carcinoma, Locally Advanced Cutaneous Melanoma, Metastatic: Acral Lentiginous Melanoma, Conjunctival Melanoma, Cutaneous Melanoma, HER2-Negative Breast Carcinoma, Mucosal Melanoma, Pathologic Stage IIIC, IIID, IV Cutaneous Melanoma, Prognostic Stage IV Breast Cancer, Recurrent: Acral Lentiginous Melanoma, Cutaneous Melanoma, Mucosal Melanoma; Refractory HER2-Negative Breast Carcinoma, Unresectable: Acral Lentiginous Melanoma, Cutaneous Melanoma, Mucosal Melanoma	NCT05098210	Recruiting	Phase 1	20	Single Arm: Neoantigen Peptide Vaccine + Nivolumab + Poly ICLC
Melanoma, Colon Cancer, Gastrointestinal Cancer, Genitourinary Cancer, Hepatocellular Cancer	NCT03480152	Terminated	Phase 1,2	5	Single Arm: NCI-4650, a messenger ribonucleic acid (mRNA)-based, Personalized Cancer Vaccine
Urothelial/Bladder Cancer	NCT03359239	Completed	Phase 1	10	Single Arm: PGV001 (multipeptide personalized neoantigen vaccine) + Atezolizumab
Anatomic Stage IV Breast Cancer, Invasive Breast Carcinoma, Metastatic Triple-Negative Breast Carcinoma	NCT03606967	Recruiting	Phase 2	70	Group 1) nab-paclitaxel + durvalumab + tremelimumab + personalized synthetic long peptide vaccine; Group 2) nab-paclitaxel + durvalumab + tremelimumab
Gastric Cancer, Hepatocellular Carcinoma, Non-Small-Cell Lung Cancer, Colon Rectal Cancer	NCT04147078	Recruiting	Phase 1	80	Single Arm: tumor neoantigen primed DC vaccine
Melanoma, Gastrointestinal Cancer, Breast Cancer, Ovarian Cancer, Pancreatic Cancer	NCT03300843	Terminated	Phase 2	1	Single Arm: Peptide loaded dendritic cell vaccine
Myeloproliferative Neoplasms	NCT05444530	Recruiting	Phase 1	60	Sequential: VAC85135 (Neoantigen Vaccine) + Ipilimumab
Malignant Melanoma, Metastatic, Non Small Cell Lung Cancer Metastatic, Bladder Urothelial Carcinoma, Metastatic	NCT03715985	Active, not recruiting	Phase 1,2	12	Sequential: EVAX-01-CAF09b (personalised NPV-ds001 drug)
Anatomic Stage III, IIIA, IIIB, IIIC, IV Breast Cancer, Clinical Stage III, IV Cutaneous Melanoma, Clinical Stage III, IV, IVA, IVB, Gastric Cancer, Clinical Stage III, IV, IVA, IVB Gastroesophageal Junction (GEJ) Adenocarcinoma, Clinical Stage III, IV Merkel Cell Carcinoma, Locally Advanced and Metastatic: Cervical, Endometrial Carcinoma, Gastric Adenocarcinoma, GEJ Adenocarcinoma, Head and Neck Squamous Cell Carcinoma, Hepatocellular Carcinoma, Lung Non-Small Cell Carcinoma, Malignant Solid Neoplasm, Melanoma, Merkel Cell Carcinoma, Renal Cell Carcinoma, Skin Squamous Cell Carcinoma, TNBC, Unresectable Breast Carcinoma, Unresectable Cervical Carcinoma,	NCT05269381	Recruiting	Phase 1	36	Single Arm: Cyclophosphamide + Personalized Neoantigen vaccine + Pembrolizumab
Unresectable Gastric Adenocarcinoma, Unresectable GEJ Adenocarcinoma, Unresectable Renal Cell Carcinoma, Urothelial Carcinoma, Pathologic Stage III: Cutaneous Melanoma, Gastric Cancer, GEJ Adenocarcinoma, Merkel Cell Carcinoma; Pathologic Stage IIIA: Cutaneous Melanoma, Gastric Cancer, GEJ Adenocarcinoma; Pathologic Stage IIIB: Cutaneous Melanoma, Gastric Cancer, GEJ Adenocarcinoma; Pathologic Stage IIIC: Cutaneous Melanoma, Gastric Cancer; Pathologic Stage IIID Cutaneous Melanoma; Pathologic Stage IV Cutaneous Melanoma, Gastric Cancer, GEJ Adenocarcinoma, Merkel Cell Carcinoma; Pathologic Stage IVA, IVB GEJ Adenocarcinoma; Postneoadjuvant Therapy Stage III, IV Gastric Cancer; Postneoadjuvant Therapy Stage III, IIIA, IIIB, IVA, IVB GEJ Adenocarcinoma, Prognostic Stage III, IIIA, IIIB, IIIC, IV Breast Cancer, Skin Squamous Cell Carcinoma; Stage III: Cervical Cancer, Cutaneous Squamous Cell Carcinoma of the Head and Neck, Hepatocellular Carcinoma, Lung Cancer, Renal Cell Cancer, Uterine Corpus Cancer; Stage IIIA Cervical Cancer, Hepatocellular Carcinoma, Lung Cancer, Uterine Corpus Cancer; Stage IIIB Cervical Cancer, Hepatocellular Carcinoma,	NCT05269381	Recruiting	Phase 1	36	Single Arm: Cyclophosphamide + Personalized Neoantigen vaccine + Pembrolizumab
Lung Cancer, Uterine Corpus Cancer; Stage IIIC Lung Cancer; Stage IIIC IIIC1 IIIC2 Uterine Corpus Cancer; Stage IV Cutaneous Squamous Cell Carcinoma of the Head and Neck, Renal Cell Cancer; Stage IV IVA IVB Cervical Cancer, Hepatocellular Carcinoma, Lung Cancer, Uterine Corpus Cancer; Triple-Negative Breast Carcinoma, Unresectable: Cervical Carcinoma, Endometrial Carcinoma, Gastric Adenocarcinoma, GEJ Adenocarcinoma, Head and Neck Squamous Cell Carcinoma, Hepatocellular Carcinoma, Lung Non-Small Cell Carcinoma, Malignant Solid Neoplasm, Melanoma, Merkel Cell Carcinoma, Renal Cell Carcinoma, Skin Squamous Cell Carcinoma, Triple-Negative Breast Carcinoma, Urothelial Carcinoma	NCT05269381	Recruiting	Phase 1	36	Single Arm: Cyclophosphamide + Personalized Neoantigen vaccine + Pembrolizumab
Hepatocellular Cancer, Colorectal Cancer, Liver Metastases	NCT04912765	Recruiting	Phase 2	60	Single Arm: Neoantigen Dendritic Cell Vaccine + Nivolumab
Biochemically Recurrent Prostate Carcinoma, Prostate Adenocarcinoma	NCT04336943	Recruiting	Phase 2	30	Single Arm: Durvalumab + Olaparib against pts with a predicted high neoantigen load
Breast Cancer, Cutaneous melanoma	NCT02831634	Completed	Not applicable	25	Not Applicable
Advanced Solid Tumor	NCT05020119	Recruiting	Phase 1	9	Sequential Neoantigen-expanded cell therapy
Endocrine Tumors, Non-Small Cell Lung Cancer, Ovarian Cancer, Breast Cancer, Gastrointestinal/Genitourinary Cancers, Neuroendocrine Tumors, Multiple Myeloma	NCT03412877	Recruiting	Phase 2	270	Group 1) cyclophosphamide + fludarabine + Individual Patient TCR-Transduced PBL + high- or low-dose aldesleukin; Group 2) cyclophosphamide + fludarabine + Individual Patient TCR-Transduced PBL + high- or low-dose aldesleukin + pembrolizumab
Melanoma	NCT01970358	Completed	Phase 1	20	Single Arm: NeoVax (peptides + poly-ICLC)
Ovarian Cancer	NCT04024878	Recruiting	Phase 1	30	Group 1) Nivolumab + NeoVax; Group 2) Nivolumab + NeoVax + Core Needle Bx upon recurrence
Melanoma	NCT04072900	Recruiting	Phase 1	30	Single Arm: Personalized NeoAntigen Cancer Vaccine- Neo-Vac-Mn (peptides + rhGM-CSF + anti-PD1 + Imiquimod 5% Topical Cream)
Pancreatic Cancer	NCT04161755	Active, not recruiting	Phase 1	29	Single Arm: Atezolizumab + RO7198457 (personalized vaccine) + mFOLFIRINOX
Glioblastoma	NCT02287428	Recruiting	Phase 1	56	Group 1) Standard Radiation (Std RT) + NeoVax; Group 2) Pembrolizumab/Std RT + NeoVax + Pembrolizumab; Group 3) Std RT + NeoVax + Pembrolizumab (Pembro); Group 4) Std RT (+ 1 dose Pembro) + NeoVax + Pembro; Group 5) Std RT + Temozolomide + NeoVax + Pembro
Lymphocytic Leukemia	NCT03219450	Recruiting	Phase 1	15	Sequential: NeoVax + Cyclophosphamide + Pembrolizumab
Follicular Lymphoma	NCT03361852	Recruiting	Phase 1	20	Single Arm: Rituximab + NeoVax + Pembro
Advanced Cancer	NCT02992977	Terminated	Phase 1	3	Single Arm: AutoSynVax™ (personalized neoantigen) vaccine
Breast Cancer Female	NCT04879888	Completed	Phase 1	9	Single Arm: Peptide pulsed Dendritic cell
Advanced Cancer	NCT03568058	Active, not recruiting	Phase 1	30	Group 1) Personalized Vaccine + Pembro; Group 2) Pembro then personalized vaccine; Group 3) Pembro + personalized vaccine; Group 4) Personalized vaccine
Solid Tumors, Adult	NCT05354323	Recruiting	Phase 1	6	Single Arm: NECVAX-NEO1 (Personalized patient-individual oral DNA vaccine)
Solid Tumor, Adult	NCT03673020	Completed	Phase 1	3	Single Arm: ASV® AGEN2017 + QS-21 Stimulon® adjuvant (Neoantigen vaccine)
High Risk Cancer, Pancreatic Cancer	NCT05013216	Recruiting	Phase 1	25	Single Arm: pooled mutant-KRAS peptide vaccine with poly-ICLC adjuvant
Cutaneous Melanoma, Non-small Cell Lung Cancer, Squamous Cell Carcinoma of the Head and Neck, Urothelial, Renal Cell Carcinoma	NCT03633110	Completed	Phase 1, 2	24	Single Arm: GEN-009 (Personalized adjuvanted vaccine) + Nivolumab + Pembro
Colorectal Cancer	NCT05238558	Active, not recruiting	Phase 1	16	Single Arm: FMPV-1 + GM-CSF (as adjuvant)
Melanoma, Non-Small-Cell Lung Carcinoma	NCT04990479	Recruiting	Phase 1	34	Single Arm: Nous-PEV (Personalized vaccine GAd-PEV + MVA-PEV)
Diffuse Intrinsic Pontine Glioma (DIPG) or Glioblastoma (GBM)	NCT03914768	Enrolling by invitation	Phase 1	10	Single Arm: Immunomodulatory DC vaccine to target DIPG and GBM
Non-Small-Cell Lung Cancer, Lung cancer, Nonsquamous nonsmall Lung Cancer	NCT03380871	Completed	Phase 1	38	Single Arm: NEO-PV-01/Adjuvant (personalized vaccine) + pembrolizumab + chemotherapy
Colorectal Cancer, Pancreatic Cancer	NCT04117087	Recruiting	Phase 1	30	Single Arm: KRAS peptide vaccine + Nivolumab + Ipilimumab
Melanoma, Non-Small Cell Lung Cancer, Bladder Cancer, Colorectal Cancer, TNBC, Renal Cancer, Head and Neck cancer, Other solid cancers	NCT03289962	Active, not recruiting	Phase 1	272	Sequential: Autogene Cevumeran + Atezolizumab
Diffuse Intrinsic Pontine Glioma, Diffuse Midline Glioma, H3 K27M- Mutant	NCT04943848	Recruiting	Phase 1	36	Sequential: rHSC-DIPGVax (neoantigen heat shock protein) + Balstilimab + Zalifrelimab
Acute Myelogenous Leukemia, Acute Lymphocytic Leukemia, Chronic Myelogenous Leukemia, Myelodysplastic Syndrome, Non-Hodgkin’s lymphoma	NCT00923910	Completed	Phase 1, 2	10	Single Arm: WT1 Peptide-Pulsed Dendritic Cells + Donor Lymphocytes + IL-4 + KLH + WT1 Peptides + Endotoxin + Diphenydramine + Acetaminophen
Metastatic: Colorectal Adenocarcinoma, Pancreatic Ductal Adenocarcinoma; Stage IV: Colorectal Cancer, Pancreatic Cancer; Stage IVA, IVB Colorectal Cancer	NCT02600949	Recruiting	Phase 1	150	Group 1) personalized vaccine, imiquimod; Group 2) personalized vaccine, imiquimod, pembrolizumab; Group 3,4) personalized vaccine, imiquimod, pembrolizumab, sotigalimab
Urinary Bladder Cancer, Transitional Cell Carcinoma of the Bladder. Malignant Melanoma, Melanoma, Skin Cancer, Non Small Cell Lung Cancer, Lung cancer	NCT02897765	Completed	Phase 1	34	Single Arm: NEO-PV-01 + Nivolumab + Adjuvant
Diffuse Intrinsic Pontine Glioma	NCT04749641	Recruiting	Phase 1	30	Group 1: Stereotactic Biopsy + Histone H3.3-K27M Neoantigen Vaccine; Group 2: Open Biopsy + Histone H3.3-K27M Neoantigen Vaccine
Glioblastoma	NCT04015700	Recruiting	Phase 1	12	Single Arm: Personalized neoantigen DNA vaccine supplied by Geneos Therapeutics + Plasmid encoded IL-12
Glioblastoma	NCT03422094	Terminated	Phase 1	3	Sequential: NeoVax + Nivolumab + Ipilimumab
Melanoma	NCT03929029	Recruiting	Phase 1	20	Single Arm: Nivolumab + NeoVax plus Montanide + Ipilimumab
Melanoma, Ocular Melanoma, Uveal Melanoma	NCT04364230	Recruiting	Phase 1, 2	44	Single Arm: 6MHP + NeoAg-mBRAF + PolyICLC + CDX-1140
Metastatic Melanoma	NCT03597282	Terminated	Phase 1	22	Group 1) NEO-PV-01 + adjuvant + nivolumab; Group 2) Nivolumab + adjuvant; Group 3) NEO-PV-01 + adjuvant + nivolumab on alternate schedule; Group 4) NEO-PV-01 + adjuvant + nivolumab + APX005M; Group 5) Nivolumab + APX005M; Group 6) NEO-PV-01 + adjuvant + nivolumab + ipilimumab; Group 7) Nivolumab + ipilimumab
Glioblastoma Multiforme of Brain	NCT04968366	Recruiting	Phase 1	10	Single Arm: Autologous dendritic cells pulsed with multiple neoantigen peptides + Temozolomide adjuvant chemotherapy
Melanoma, Metastatic Melanoma	NCT04930783	Recruiting	Phase 1	20	Single Arm: NEOVAX + CDX-301 + Nivolumab
Glioblastoma Multiforme, Astrocytoma, Grade IV	NCT02510950	Terminated	Phase 1	1	Single Arm: Personalized peptide vaccine + Poly-ICLC + Temozolomide
Malignant, Recurrent Glioma	NCT04943718	Recruiting	Phase 1	10	Single Arm: personalized vaccine

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
