# Peer review of "The Identification and Clinical Applications of Mutated Antigens in the Era of Immunotherapy"

_cancers, 2022, doi:10.3390/cancers14174255_

Round 1
Reviewer 1 Report
In this review by Kye et al., the authors discuss approaches to target neoantigens for cancer immunotherapy. While this review is important and timely, re-organization of some sections would enhance the manuscript and inclusion of clinical results would highlight the importance of these strategies.
The introduction should be re-organized and clarified. Perhaps after a description of how T cells (maybe include CD4+ T cells in addition to CD8+ T cells) are activated, then neoantigens could be discussed. Then a brief paragraph introducing the approaches to targeting neoantigens that will be discussed in the review. As it stands, paragraph 3 is confusing, the paragraph about monoclonal antibodies should be re-worked to link monoclonal antibodies to neoantigens, and the discussion about MHC loss and NK cell activity may be better suited to the limitations paragraph at the end of the review.
Figure 1 is too small.
Figure Legend 1 is missing the description of the “Monoclonal Antibodies for Immunotherapy”. This Figure may need to be revised when the Introduction is revised.
Line 59, this sentence is unclear and could be reworded. For example, “However, in order to prevent autoimmunity, responses are often not potent enough…”
In Section 2, very little attention is given to CTAs or how they are aberrantly expressed on tumors. A discussion of why CTAs are not optimal antigens could be included. It seems like this section should just discuss different tumor antigens and their benefits/challenges, while the vaccine discussion should be part of section 5. Monoclonal antibodies could be separated out and included in its own section. A discussion of how neoantigens are identified could be included here too.
In Section 2, the discussion of monoclonal antibodies does not really encompass antibodies against neoantigens as most current therapies target overexpressed normal proteins, such as HER2. Also, the authors should remind the reader that most monoclonal antibodies (other than TCRm) will recognize antigens on the surface of tumor cells.
In Section 3, the sentence on lines 105-106 is confusing since TIL are not engineered to recognize TAAs. In addition, line 108 makes no sense as it is clear that TIL recognize tumors based on MHC.
In Section 4, line 123 should be reworded to clarify that T cells are genetically modified to express TCRs that recognize tumor-specific epitopes. A brief description of how neoantigen-specific TCRs are discovered would be helpful. Results from previous clinical trials or case reports could be reported.
In Section 5, the last sentence needs to be clarified as the creation of activated T cells is not used to formulate the vaccines. Also, Figure 2 seems to show a neoantigen T cell therapy, not a vaccine therapy. Results from completed trials could be included.
Table 1 should be revised to exclude the studies in which no neoantigen therapeutic vaccine / T cell therapy is given. The table could also be re-organized to group the vaccine studies vs the T cell studies.
Reviewer 2 Report
In this review, Kye et al. discussed various cancer immunotherapies strategies emphasizing Neoantigens. I find the review article to be promising. However, before considering this review for publication, I highly recommend the following modifications.
1- Line 20, the sentence "various cancer cells have the ability to release tumor antigens that" makes it sound like the tumor cells require the immune response against them. Rephrase the sentence.
2- Line 30, "this provides Natural Killer (NK) cells to take part in the anti-tumor immune response," Rephrase due to grammatical error.
3- Line 45: "The generation and utilization of monoclonal antibodies for 45 immunotherapy are: 1)"., Rephrase.
4- In Section 2 (Starting from Line 55), it will be beneficial to introduce the various types of tumor antigens before writing about the limitations.
5- In Line 100 ICI is written without the full name. The full name is introduced in the following line only. Correct the mistake.
6- Line 123 include the word, CAR-T therapy without any previous discussion about it. It is crucial to add a section for CAR-T Therapy in this manuscript.
7- Line 125-126 rephrase.
8- Figure 2 - Writing Neoantigen Vaccination is confusing. I assume that part of the figure was meant to represent the steps' beginning and end. To make it more straightforward, indicate the biopsy collection in that part rather than separately.
9- I recommend another figure, in which sections 2-5 are summarized in a figure format (Including CAR-T as well). This figure will be highly beneficial for the readers.
Round 2
Reviewer 1 Report
The authors have addressed the previous comments, but additional clarifications are needed.
Figure 1D: The description of CAR T cells in the figure legend needs clarification: “Isolate T cells from the patient and mount a synthetic TCR onto the host T cells”. “TCR” should be replaced with “CAR construct”.
The authors describe "neoantigen-specific TCR-T cell therapy" but the text and figure 3A describes an enriched TIL product, not TCR transduced T cells. This should be clarified in the text/figure. Additional details to describe neoantigen-specific TCR-transduced T cells should be mentioned in this section too.
Author Response
Reviewer Comment: Figure 1D: The description of CAR T cells in the figure legend needs clarification: “Isolate T cells from the patient and mount a synthetic TCR onto the host T cells.” “TCR” should be replaced with “CAR construct.”
Author’s Response: Thank you for pointing this out. We have modified the figure legend to read
“Isolate T cells from the patient and mount a synthetic CAR construct onto the host T cells.”
Figure 1D has also been updated to reflect this change.
Reviewer Comment: The authors describe "neoantigen-specific TCR-T cell therapy" but the text and figure 3A describes an enriched TIL product, not TCR transduced T cells. This should be clarified in the text/figure. Additional details to describe neoantigen-specific TCR-transduced T cells should be mentioned in this section too.
Author’s Response: We agree with the reviewer’s comment. We have modified Figure 3A to be reflective of enriched autologous neoantigen specific T cells. This section has been modified to eliminate the word TIL from the section and focus on TCR T transdued cells.
Reviewer 2 Report
The manuscript is highly improved. The figures summarize the sections very well. The review in its current version provides an excellent summary and good background for readers interested in Immunotherapies for Cancer.
The current version of Table 1 is missing a header (Columns don't have titles). After correcting that, I recommend publishing this review.
Author Response
Reviewer 2. The manuscript is highly improved. The figures summarize the sections very well. The review in its current version provides an excellent summary and good background for readers interested in Immunotherapies for Cancer.
Reviewer Comment: The current version of Table 1 is missing a header (Columns don't have titles). After correcting that, I recommend publishing this review.
Author’s Response: We have added headers (column titles) to each new page of the tables. We hope this provides more clarity when reviewing the table figure.